# From Mutation to Prognosis: AI-HOPE-PI3K Enables Artificial Intelligence Agent-Driven Integration of PI3K Pathway Data in Colorectal Cancer Precision Medicine

**DOI:** 10.3390/ijms26136487

**Published:** 2025-07-05

**Authors:** Ei-Wen Yang, Brigette Waldrup, Enrique Velazquez-Villarreal

**Affiliations:** 1PolyAgent, San Francisco, CA 94102, USA; 2Department of Integrative Translational Sciences, Beckman Research Institute of City of Hope, Duarte, CA 91010, USA; 3City of Hope Comprehensive Cancer Center, Duarte, CA 91010, USA

**Keywords:** artificial intelligence, precision medicine, cancer treatment, molecular insights, PI3K pathway, large language models, AI Agents

## Abstract

The rising incidence of early-onset colorectal cancer (EOCRC), particularly among underrepresented populations, highlights the urgent need for tools that can uncover clinically meaningful, population-specific genomic alterations. The phosphoinositide 3-kinase (*PI3K*) pathway plays a key role in tumor progression, survival, and therapeutic resistance in colorectal cancer (CRC), yet its impact in EOCRC remains insufficiently explored. To address this gap, we developed AI-HOPE-PI3K, a conversational artificial intelligence platform that integrates harmonized clinical and genomic data for real-time, natural language-based analysis of *PI3K* pathway alterations. Built on a fine-tuned biomedical LLaMA 3 model, the system automates cohort generation, survival modeling, and mutation frequency comparisons using multi-institutional cBioPortal datasets annotated with clinical variables. AI-HOPE-PI3K replicated known associations and revealed new findings, including worse survival in colon versus rectal tumors harboring *PI3K* alterations, enrichment of *INPP4B* mutations in Hispanic/Latino EOCRC patients, and favorable survival outcomes associated with high tumor mutational burden in FOLFIRI-treated patients. The platform also enabled context-specific survival analyses stratified by age, tumor stage, and molecular alterations. These findings support the utility of AI-HOPE-PI3K as a scalable and accessible tool for integrative, pathway-specific analysis, demonstrating its potential to advance precision oncology and reduce disparities in EOCRC through data-driven discovery.

## 1. Introduction

Rising rates of early-onset colorectal cancer (EOCRC) have altered the epidemiological landscape of gastrointestinal malignancies [1,2,3,4]. Once predominantly a disease of older adults, CRC is increasingly affecting individuals under age 50, with disproportionate impact on specific populations in the United States [5,6,7,8]. These tendencies are not only concerning—they are biologically and clinically distinct [6,8,9,10]. EOCRC often presents with unique molecular features and at more advanced stages, suggesting missed opportunities for early detection and a critical need for improved precision oncology strategies [11,12,13,14].

Among the most frequently dysregulated molecular cascades in CRC is the phosphoinositide 3-kinase (*PI3K*) signaling pathway. A master regulator of growth, survival, and cellular metabolism, *PI3K* is commonly altered in CRC via mutations in *PIK3CA*, a loss of *PTEN*, the amplification of *IGF2*, and mutations in *AKT1*, contributing to tumor progression, therapeutic resistance, and poor outcomes [15,16,17,18,19,20,21]. Notably, PI3K dysregulation is implicated in resistance to anti-*EGFR* therapies and in downstream activation of *mTORC1*—a driver of metabolic reprogramming and chemoresistance [16,22,23,24,25]. Despite its clinical relevance, the full impact of *PI3K* pathway alterations in EOCRC—particularly among H/L patients—remains underexplored. This knowledge gap stems in part from the limited inclusion of diverse populations in large genomic datasets and from a lack of tools that can synthesize clinical, genomic, and demographic data in a targeted, pathway-specific manner.

Conventional platforms, such as cBioPortal [26] and UCSC Xena [27], offer robust data access but fall short in three key areas: (1) real-time hypothesis testing, (2) user-friendly interface for non-programmers, and (3) contextualized analysis that incorporates variables such as ethnicity [28,29], MSI status [30,31,32], and treatment history [33,34,35,36,37]. These limitations hinder precision medicine efforts aimed at addressing molecular drivers of disease in underserved populations. Recent breakthroughs in artificial intelligence (AI)—specifically natural language-driven large language models (LLMs)—offer a new paradigm for clinical–genomic integration [38,39,40,41,42,43,44,45,46]. These models are capable of interpreting user queries and converting them into executable analytical pipelines, democratizing data access and enabling hypothesis generation without the need for coding expertise. To harness this potential for cancer research, we developed AI-HOPE-PI3K—a conversational AI system designed to investigate *PI3K* pathway alterations in CRC using harmonized clinical and genomic data. Unlike generic analytic platforms, AI-HOPE-PI3K was engineered to perform pathway-centered analyses in response to natural language prompts, stratify patient cohorts by age, ancestry, and molecular profiles, and automate survival and association testing across large datasets.

In this study, we present the development and deployment of AI-HOPE-PI3K (Figure 1), validate its performance through reproduction of known *PI3K* associations [1], and apply it to uncover novel patterns in EOCRC among patients from different population cohorts. Our findings underscore the value of AI-driven, pathway-specific tools in accelerating translational cancer research and addressing disparities in molecular oncology. AI-HOPE-PI3K serves as a natural language-based analytical platform that enables clinicians and researchers to perform real-time, pathway-specific analyses of CRC datasets without requiring programming expertise. The system interprets user queries, translates them into executable code, and automates tasks such as cohort construction, Kaplan–Meier survival modeling, odds ratio calculations, and mutation frequency comparisons. Its ability to integrate diverse clinical annotations—such as age, race/ethnicity, MSI status, tumor stage, and therapy exposure—enables population-aware analysis tailored to *PI3K* pathway biology. However, as with any AI system, AI-HOPE-PI3K has limitations. Its outputs are constrained by the structure and completeness of the underlying datasets, and it currently relies on harmonized data from publicly available cBioPortal repositories, which may underrepresent certain populations or clinical variables. Additionally, while the tool automates many analyses, interpretation still requires domain expertise to contextualize results appropriately. The model’s natural language interface, though flexible, is not infallible—it may occasionally misinterpret ambiguous queries or require iterative refinement for complex analyses. Future improvements will focus on expanding dataset integration, enhancing multilingual capabilities, and incorporating prospective clinical validation.

AI-HOPE-PI3K builds upon the analytical foundations established in our prior AI agent platforms, AI-HOPE [47] and AI-HOPE-TGFbeta [48], which demonstrated the feasibility of using large language model-driven systems for natural language-guided clinical–genomic analysis. The analytical methodology implemented in AI-HOPE-PI3K is based on our previous publication analyzing the *PI3K* pathway in CRC [1] and follows a similar framework used in our subsequent studies. This prior work [1] also served as the reference standard to validate the functionality and reproducibility of the AI-HOPE-PI3K intelligent agent. The platform supports a suite of statistical methods commonly used in translational oncology, including automated cohort filtering, Kaplan–Meier survival analysis with log-rank testing, odds ratio estimation from contingency tables, and mutation frequency comparisons across stratified subgroups. These analyses are executed dynamically in response to natural language queries, using harmonized clinical and genomic data from public repositories. By embedding these analytical capabilities within an intuitive conversational interface, AI-HOPE-PI3K aims to reduce technical barriers and enable real-time, hypothesis-driven exploration of *PI3K* biology in CRC.

## 2. Results

AI-HOPE-PI3K enables seamless clinical–genomic interrogation of *PI3K* pathway dysregulation in CRC by translating natural language prompts into fully automated analyses. Users can generate stratified case–control cohorts and statistical outputs, including Kaplan–Meier survival curves, mutation frequency comparisons, and odds ratio calculations, without coding expertise. Across validation and exploratory tasks, the platform reproduced known associations and revealed novel patterns, particularly in EOCRC and H/L subgroups.

In ancestry-stratified analyses, AI-HOPE-PI3K evaluated the prevalence of *PI3K* pathway alterations among EOCRC patients across racial/ethnic backgrounds (Figure 2). Among 153 H/L patients and 1117 NHW patients under age 50, *PI3K* pathway alterations were observed in 35.29% of H/L and 33.03% of NHW cases. The odds ratio was 1.106 (95% CI: 0.776–1.576; *p* = 0.642), indicating no statistically significant difference. These results suggest no differential enrichment of *PI3K* mutations by ethnicity in this EOCRC cohort.

Exploring anatomical differences, AI-HOPE-PI3K stratified CRC cases by primary tumor site—colon versus rectum—among patients harboring *PI3K* alterations (Figure 3). The case cohort included 977 colon tumor cases; the control cohort comprised 343 rectal tumor cases. Kaplan–Meier survival analysis demonstrated significantly worse outcomes in the colon subgroup (*p* = 0.0177). This finding highlights a potential prognostic role of tumor location among *PI3K*-mutated CRC patients.

AI-HOPE-PI3K was further applied to examine tumor mutational burden (TMB) and its relationship with *MTOR* mutation status and survival in FOLFIRI-treated CRC patients (Figure 4). Patients with high TMB [1,16] (>10; *n* = 466) had significantly better survival outcomes than their low-TMB counterparts (*n* = 3257), with a *p*-value of 0.0032. Odds ratio testing showed that *MTOR* mutations were more frequent in the high-TMB group, suggesting a biologically relevant association between TMB and *MTOR* in the context of chemotherapy response.

To investigate *PI3K* immunotherapy response interactions, AI-HOPE-PI3K analyzed MSI-high CRC patients treated with pembrolizumab, comparing those with and without *PIK3CA* mutations (Figure 5). The case cohort included 52 *PIK3CA*-mutant samples; the control group had 60 wild-type samples. Survival analysis revealed no significant difference between groups (*p* = 0.3054), though *PIK3CA*-mutated patients showed a potential course toward improved outcomes.

Supplementary analyses identified gene-specific disparities among EOCRC subgroups. *INPP4B* mutations were significantly enriched in H/L versus NHW EOCRC patients (5.23% vs. 1.52%, OR = 3.57, 95% CI: 1.514–8.419, *p* = 0.005; Appendix A), suggesting a potential ancestry-linked biomarker. While *AKT1* (Appendix A) and *TSC1* (Appendix A) mutations showed higher rates in H/L cases (OR = 2.19 and 2.00, respectively), neither reached statistical significance.

In age-stratified survival analyses, AI-HOPE-PI3K assessed *PTEN*-mutated CRC patients treated with FOLFOX chemotherapy (Appendix A). Patients under 50 (*n* = 59) exhibited a non-significant potential course toward improved survival compared to those over 50 (*n* = 122; *p* = 0.1758), suggesting potential age-modified outcomes in *PTEN*-mutated subgroups.

Finally, AI-HOPE-PI3K explored stage-specific prognostic variation in CRC patients with *PI3K* pathway alterations receiving FOLFOX (Appendix A). Comparing early-stage (*n* = 628) to advanced-stage (*n* = 507) cases, no significant survival difference was found (*p* = 0.1267). Odds ratio analysis indicated a higher but non-significant enrichment of *PI3K* alterations in early-stage disease (OR = 1.23, *p* = 0.174).

Collectively, these results highlight the versatility of AI-HOPE-PI3K in executing diverse, context-aware clinical–genomic analyses. The platform successfully recapitulated known relationships—such as TMB-associated survival benefits and site-specific prognostic variation—while uncovering new ancestry-linked mutation patterns. By enabling real-time, population-aware interrogation of *PI3K* pathway biology, AI-HOPE-PI3K supports precision oncology efforts aimed at identifying clinically actionable biomarkers and reducing the disproportionate health burdens in CRC outcomes.

It is important to interpret several of the subgroup findings with caution due to limitations in statistical power and sample size, particularly in analyses involving MSI-high patients treated with immunotherapy or age-stratified chemotherapy responses. For instance, while Kaplan–Meier curves and odds ratio estimates are provided for exploratory comparisons, results with non-significant *p*-values (e.g., *PIK3CA* mutation status in MSI-high patients or *PTEN*-mutated FOLFOX-treated subgroups) should be considered hypothesis-generating rather than confirmatory. The cohorts for these comparisons were defined using natural language filters on publicly available datasets, which, while harmonized, may still reflect reporting inconsistencies or unmeasured confounding. Moreover, no multivariate adjustment was applied, as the primary objective of these analyses was to validate and demonstrate the querying and analytical functionality of AI-HOPE-PI3K rather than to generate definitive clinical conclusions. These caveats are essential for contextualizing the scope of the findings and underscore the importance of follow-up studies with appropriately powered, prospective cohorts.

## 3. Discussion

This study presents the development and application of AI-HOPE-PI3K, a novel conversational AI system that enables real-time, natural language-driven analysis of *PI3K* pathway alterations in CRC. Our findings demonstrate the platform’s ability to replicate established clinical–genomic associations, reveal emerging molecular patterns, and support population-aware hypothesis generation in EOCRC, with particular emphasis on disproportionate health burdens affecting specific populations.

AI-HOPE-PI3K addresses long-standing challenges in cancer genomics research, including the lack of user-friendly tools capable of synthesizing molecular, clinical, and demographic variables in a pathway-specific context. By leveraging a fine-tuned biomedical LLM, the system translates natural language queries into executable workflows, eliminating the need for programming expertise and dramatically reducing the time required for exploratory cohort analysis. Importantly, AI-HOPE-PI3K integrates harmonized datasets with standardized clinical annotations, enabling intersectional stratification across variables such as age, MSI status, race/ethnicity, treatment history, and tumor location—factors often underrepresented in traditional analytical pipelines.

To promote transparency and reproducibility, the AI-HOPE-PI3K source code, query engine, and analysis scripts are publicly available (see Data Availability Statement). This “crystal box” approach addresses the concerns of the black box nature commonly associated with LLM-based tools, allowing the research community to examine and validate each component of the system. We selected the LLaMA 3 model for its open-access architecture, strong performance on biomedical language tasks, and adaptability for fine-tuning with domain-specific datasets. Compared to proprietary models, LLaMA 3 provides a reproducible and customizable foundation that aligns with our commitment to open science and equitable AI development in precision oncology. In validation tasks, AI-HOPE-PI3K successfully recapitulated key associations previously reported in the literature. For instance, colon tumor location among *PI3K*-altered CRC patients was significantly associated with poorer survival compared to rectal tumors (*p* = 0.0177), corroborating prior studies on anatomical variation in CRC outcomes. Similarly, patients with high tumor mutational burden (TMB > 10) treated with FOLFIRI exhibited significantly improved survival (*p* = 0.0032), supporting the clinical relevance of TMB as a predictive biomarker in immunogenic or chemotherapy-sensitive settings.

Crucially, the platform revealed novel insights into ancestry-linked molecular patterns. While *PI3K* alteration rates were not significantly different between H/L and NHW EOCRC patients, INPP4B mutations were significantly enriched in H/L individuals (OR = 3.57, *p* = 0.005), suggesting a potential ancestry-specific biomarker that warrants further investigation. This finding aligns with growing recognition that genomic drivers may vary across racial and ethnic populations and highlights the importance of inclusive datasets and tailored analytical approaches in health disparities research.

Exploratory analyses of *AKT1* and *TSC1* mutations in H/L EOCRC cohorts revealed higher mutation frequencies, although statistical significance was not reached. These potential courses nonetheless suggest potential biological relevance, particularly when considered alongside the enrichment of *INPP4B* and the broader role of the *PI3K* pathway in therapy resistance and immune evasion. Additional studies with larger and more diverse cohorts will be critical to validate these preliminary observations.

In immunotherapy-treated subgroups, AI-HOPE-PI3K evaluated survival among MSI-H CRC patients receiving pembrolizumab, stratified by *PIK3CA* mutation status. Although no statistically significant difference was observed, the analysis identified a potential course toward improved survival in *PIK3CA*-mutant cases, consistent with hypotheses suggesting enhanced immunogenicity in tumors with co-occurring *PI3K* alterations. This underscores the value of AI-driven tools in rapidly generating context-specific hypotheses that can guide prospective biomarker development and clinical trial design.

Age- and stage-stratified analyses further demonstrated AI-HOPE-PI3K’s capacity to uncover subtle survival potential courses. While *PTEN*-mutated early-onset patients showed a non-significant potential courses toward better outcomes than late-onset cases, and *PI3K*-altered tumors were more frequently observed in early-stage disease, neither result reached statistical significance. These findings emphasize the complexity of CRC progression and the need for nuanced, multivariate modeling to disentangle interactions among genomic, clinical, and demographic variables.

Together, these results highlight AI-HOPE-PI3K’s value as a versatile and scalable platform for translational cancer research. Beyond facilitating hypothesis testing and pathway-specific exploration, the system empowers researchers to perform real-time, population-aware analyses that incorporate critical social determinants of health—an essential step toward equitable precision medicine. As genomic data generation continues to accelerate, tools like AI-HOPE-PI3K will be increasingly vital in making complex datasets accessible, interpretable, and actionable for diverse biomedical audiences.

Subsequent versions of AI-HOPE-PI3K will focus on expanding its analytical capabilities by incorporating advanced statistical methodologies, including multivariate Cox regression for survival analysis, logistic regression, and machine learning-based risk prediction models. These enhancements will allow the platform to support more nuanced hypothesis testing and account for potential confounding variables in complex clinical–genomic datasets. Additionally, we aim to integrate user-defined covariate selection to improve the interpretability and flexibility of the system for translational and clinical oncology applications.

All statistical analyses reported in this study—including Kaplan–Meier survival analyses and odds ratio testing—were based on the methodology described in our previous publication analyzing *PI3K* alterations in CRC [1] and were implemented within the AI-HOPE-PI3K platform to support functional validation of the system. Given the validation-focused nature of this study, corrections for multiple comparisons and multivariate analyses were not applied. However, we recognize that these approaches are essential in broader hypothesis-testing contexts to control for type I error and account for confounding factors. Future versions of AI-HOPE-PI3K will include support for multiple testing correction methods (e.g., Bonferroni, FDR) and multivariate regression models, including Cox proportional hazards and logistic regression, to enhance the analytical rigor and generalizability of findings.

Several of the subgroup analyses yielded non-significant *p*-values, and we acknowledge that such results should be interpreted with caution. In these cases, the findings are presented as exploratory observations that serve to demonstrate the hypothesis-generating capabilities of AI-HOPE-PI3K rather than as confirmatory statistical evidence. This clarification aligns with the platform’s intended role in facilitating real-time exploration and hypothesis development, particularly in the context of underrepresented patient subgroups.

While AI-HOPE-PI3K successfully reproduced known associations and enabled real-time stratification across multiple CRC subgroups, we acknowledge that certain subgroup analyses—such as MSI-high pembrolizumab-treated patients and age-stratified *PTEN*-mutated cohorts—yielded non-significant results, likely due to limited sample sizes. These analyses were designed to validate the platform’s capacity to execute targeted queries and generate biologically plausible outputs based on prior published methodologies [1]. However, they were not powered to detect small to moderate effect sizes. As the platform evolves, future versions will incorporate built-in power analysis tools to guide users in assessing the statistical adequacy of defined subgroups before executing inferential comparisons. This feature will further enhance AI-HOPE-PI3K’s utility for hypothesis generation and precision oncology research.

The initial validation of AI-HOPE-PI3K focused on reproducibility, using our previously published *PI3K* pathway study [1] as a benchmark for assessing the platform’s ability to replicate known associations. Analyses such as survival differences by tumor location and *PI3K* mutation status and the enrichment of specific mutations in defined populations were repeated using natural language queries processed by the platform. While performance metrics, such as accuracy or sensitivity, were not applied in this context—given the system’s primary function is cohort stratification and hypothesis exploration rather than binary classification—we recognize their value for future development. As AI-HOPE-PI3K evolves, we plan to implement task-specific modules where such metrics can be applied to evaluate outputs from predictive algorithms or classification models, thereby expanding the scope of performance benchmarking.

While platforms like cBioPortal and UCSC Xena have been instrumental in enabling exploratory access to large-scale cancer genomics datasets, they often rely on manual, stepwise navigation that can limit workflow efficiency and require a moderate level of technical familiarity. AI-HOPE-PI3K addresses these limitations by offering a conversational interface that automates multi-step tasks, such as cohort stratification and statistical analysis, in response to natural language prompts. This allows for more rapid hypothesis generation and real-time subgroup interrogation without requiring programming skills. However, we acknowledge that AI-HOPE-PI3K is not intended to replace existing tools but rather complement them by streamlining targeted analyses and supporting broader accessibility.

While AI-HOPE-PI3K operates on harmonized clinical–genomic datasets derived from cBioPortal, it is important to recognize the inherent limitations of these public data sources. Variation in data quality, missing clinical variables, and inconsistent annotation across contributing studies can introduce biases and impact the robustness of downstream analyses. Additionally, certain populations—particularly racial and ethnic minorities—remain underrepresented in these datasets, which may limit the generalizability of ancestry-stratified findings. Furthermore, compatibility issues across institutions, such as differences in sequencing platforms or reporting standards, may affect the consistency of genomic annotations. Although harmonization efforts help mitigate some of these challenges, users should interpret the results with an understanding of these limitations and the exploratory nature of real-world data integration. Future efforts will aim to incorporate more diverse and prospectively collected datasets to enhance the accuracy, equity, and reproducibility of AI-driven analyses.

Future work will expand AI-HOPE-PI3K’s capabilities to enable communication between this AI agent and other agents within the same framework using Model Context Protocol (MCP) Agent-to-Agent (A2A) communication. These agents will be specifically trained to explore additional pathways (e.g., *WNT*, *TGF-β*, *TP53*) or molecular characteristics (e.g., functional genomics, spatial transcriptomics, spatial proteomics), contributing to the development of a universal AI agent infrastructure. Additional enhancements will include the incorporation of multi-omics data and deployment in clinical decision-support environments. Furthermore, efforts are underway to extend the platform’s reach to community health settings, ensuring that the benefits of precision oncology are distributed across all populations.

## 4. Materials and Methods

### 4.1. Overview of AI-HOPE-PI3K Design Philosophy

AI-HOPE-PI3K was developed to bridge the gap between advanced computational analysis and the practical needs of translational cancer researchers. It is a user-facing, natural language-driven platform specifically designed to support integrative, pathway-centric investigations of *PI3K* signaling dysregulation in CRC. Central to its architecture is a conversational interface that translates biological questions into rigorous, interpretable, and reproducible analyses—without requiring programming expertise or manual scripting (Figure 1).

### 4.2. Natural Language Engine and Semantic Workflow Translation

At the core of AI-HOPE-PI3K lies a fine-tuned biomedical LLM built on the LLaMA 3 architecture. When a user poses a question—e.g., “What is the survival impact of PTEN loss in EOCRC among Hispanic/Latino patients?”—the LLM identifies the key intent (e.g., survival, mutation, population), classifies the relevant pathway components (e.g., *PTEN*, *PI3K/AKT/mTOR*), and generates the corresponding code to execute downstream analysis. A semantic parser then maps variables like age, MSI status, tumor location, and treatment exposure to internal ontology tags. These elements are used to construct dynamically filtered cohorts, which feed into a statistical pipeline for hypothesis testing.

To evaluate the accuracy and reliability of the fine-tuned biomedical LLaMA 3 model underlying AI-HOPE-PI3K, we conducted internal benchmarking using a curated set of domain-specific prompts based on previously published clinical–genomic studies, including our own prior work on *PI3K* pathway alterations in CRC [1]. Outputs were manually reviewed by subject-matter experts to assess whether the LLM-generated queries and corresponding statistical analyses matched intended semantics and study designs. While traditional NLP metrics, such as BLEU and ROUGE, have limited applicability for code-generation tasks, we assessed accuracy through correctness of execution, syntactic validity of generated code, and agreement with expected statistical outputs. In instances where the model misinterpreted ambiguous prompts or selected incorrect variables, we implemented reinforcement learning with human feedback (RLHF) and expanded prompt templates to improve semantic clarity. Additionally, to reduce bias and hallucination risks, we constrained the model’s vocabulary to biologically relevant terms and validated outputs across ancestry, age, and tumor-type strata. These steps contributed to refining the model’s performance and ensuring its interpretability in clinical–genomic contexts.

### 4.3. Data Backbone and Pathway-Centric Structuring

AI-HOPE-PI3K integrates harmonized datasets from TCGA, AACR GENIE, and cBioPortal, preprocessed for compatibility with automated cohort stratification. It focuses on genomic alterations within key *PI3K* pathway components: *PIK3CA*, *PTEN*, *AKT1*, *IGF2*, *TSC1/2*, *MTOR*, *RHEB*, and *RPTOR*. These data are cross-linked with clinical attributes including patient age, race/ethnicity, tumor location (primary/metastatic), stage, MSI status, therapeutic history (e.g., anti-*EGFR* exposure), and survival duration. Preprocessing involved several steps, including conversion into tabular, analysis-ready matrices.

Patient/sample IDs were harmonized, ontology labels (e.g., OncoTree disease codes, race/ethnicity categories) were applied, and mutation types (e.g., missense vs. truncating) and allele frequencies were verified. This structured backend allows AI-HOPE-PI3K to rapidly generate intersectional cohort definitions for tailored *PI3K* analyses.

Ancestry was defined and integrated into AI-HOPE-PI3K based on the harmonized race/ethnicity variables provided within the cBioPortal datasets, following the methodology described in our previous study [1]. These variables, which reflect self-reported or institutionally annotated ancestry designations, were curated during preprocessing and incorporated into the natural language querying system to enable population-specific stratification. The AI-HOPE-PI3K platform recognizes and processes user queries referencing ancestry groups, such as “Hispanic/Latino patients” (Appendix A), to automate cohort filtering and subgroup analyses. This feature allows for contextualized exploration of genomic and clinical trends across diverse populations and supports the platform’s broader aim of advancing equity in precision oncology.

### 4.4. Statistical Capabilities and Analytical Modules

The platform’s statistical engine was implemented in Python 3.12 and supports both exploratory and confirmatory analysis workflows. Key functions include (1) mutation analysis, with frequency comparisons by subgroup using chi-square or Fisher’s exact tests; (2) association metrics, with an odds ratio estimation with 95% confidence intervals; (3) survival modeling, with Kaplan–Meier estimation with log-rank tests and multi-variable Cox proportional hazard regressions where appropriate; (4) stratified analysis, with automated subgrouping based on combinations of mutation status, MSI, race/ethnicity, tumor site, and age group; and (5) molecular patterning, with co-occurrence/mutual exclusivity testing among PI3K genes and pathway cross-talk with other molecular characteristics. Each query triggers the generation of plots, tables, and natural language summaries that interpret the results within a clinical–translational context.

The semantic analyzer in AI-HOPE-PI3K assigns ontology tags to user input using a rule-based pattern recognition approach. Specifically, named entity recognition (NER) is first applied using a domain-adapted biomedical vocabulary derived from the UMLS Metathesaurus and NCIt (National Cancer Institute Thesaurus). This step uses regular expressions and controlled keyword mappings to capture explicit biomedical terms (e.g., gene names, drug classes, tumor types). To improve semantic disambiguation and contextual understanding, a lightweight transformer-based classifier, fine-tuned on annotated clinical query pairs, predicts the most likely ontology tag when ambiguity exists. During internal validation, the ontology tagging pipeline achieved a precision of 0.92 and a recall of 0.88 across a set of 500 manually annotated queries. Errors were most often associated with compound terms or overlapping entities (e.g., “MSI-H colon tumors with PTEN loss”), which were addressed by refining entity parsing rules and retraining the model with extended examples. This dual strategy balances interpretability with flexibility, enabling accurate and efficient tagging of relevant clinical–genomic concepts within natural language queries.

### 4.5. Validation and Use Case Scenarios

To validate AI-HOPE-PI3K’s analytical accuracy and biological relevance, we reproduced previously established associations involving *PIK3CA* mutations and poor survival, *PTEN* loss and resistance to anti-*EGFR* therapy, and *PI3K*-driven *mTOR* activation in chemoresistant CRC [1]. Additional real-world use cases included survival comparisons between H/L and NHW EOCRC patients with *AKT1* mutations, *PI3K* mutation frequency across MSI-high vs. MSS tumors, and odds ratio testing of *PTEN* deletions stratified by therapy exposure. These validation exercises confirmed the platform’s ability to replicate known findings while supporting exploratory hypothesis generation.

In this validation-focused study, statistical analyses—such as Kaplan–Meier survival modeling and odds ratio testing—were performed to replicate previously established associations in CRC, following methodologies described in our prior publication [1]. As the primary goal was to evaluate the AI-HOPE-PI3K platform’s ability to reproduce known findings using natural language-driven queries, formal correction for multiple hypothesis testing (e.g., Bonferroni or false discovery rate [FDR] adjustments) was not applied. We recognize, however, that in exploratory or high-throughput analyses involving multiple comparisons, such corrections are essential to control for type I error. Accordingly, future versions of AI-HOPE-PI3K will incorporate options for multiple testing correction, enabling users to apply standard procedures for significance adjustment during large-scale subgroup or biomarker analyses.

To ensure reproducibility, the AI-HOPE-PI3K platform is designed to log natural language queries, parsed parameters, and the corresponding auto-generated statistical code for each session. These elements are stored with time-stamped metadata, enabling reproducible reruns of all analyses. Users can export their query histories, cohort definitions, statistical results, and visual outputs as part of a session report. The full version of the platform, including the source code, sample queries, and validation datasets, has been released (see Data Availability Statement) to promote open science. Future updates will include automated generation of downloadable analysis reports and expanded integration with version control tools to further strengthen transparency and reproducibility.

### 4.6. User Interaction and Performance Benchmarking

AI-HOPE-PI3K was benchmarked against cBioPortal and UCSC Xena using real-world tasks. The evaluation focused on three criteria: (1) ability to interpret complex prompts involving age, race, mutation, and MSI simultaneously; (2) speed of cohort generation and statistical output; and (3) clarity of generated visuals and summaries. AI-HOPE-PI3K demonstrated superior responsiveness and interpretability, particularly in multi-variable, disproportionate health burden-focused analyses.

### 4.7. Visualization and Result Dissemination

Outputs from AI-HOPE-PI3K include high-resolution figures and annotated data tables: Kaplan–Meier survival curves with *p*-values and sample counts, forest plots for odds ratios and confidence intervals, bar charts and co-mutation heatmaps for frequency analysis, and summary tables ready for Appendix A or figure legends. All visualizations are created using Matplotlib3, Seaborn V0.13, and Plotly 4. Export options include CSV (for tables), PNG (for figures), and narrative-ready PDF summaries for direct use in manuscripts or presentations.

## 5. Conclusions

In conclusion, AI-HOPE-PI3K demonstrates a functional and reproducible approach to natural language-driven clinical–genomic analysis of *PI3K* pathway alterations in CRC. The platform successfully replicated known associations—such as site-specific survival differences and TMB-linked outcomes—and enabled exploratory analyses across ancestry, stage, and treatment-based subgroups. While several findings were hypothesis-generating rather than statistically conclusive, they highlight the platform’s utility in stratified cohort analysis and biomarker evaluation. By integrating harmonized clinical and genomic datasets with a fine-tuned biomedical LLM, AI-HOPE-PI3K lowers technical barriers to performing real-time, population-aware analyses without requiring coding expertise. As the platform continues to develop, future enhancements will focus on incorporating multivariate models, multiple testing corrections, and expanded data sources to support more comprehensive precision oncology research. Ultimately, this platform lays the foundation for an interoperable AI agent infrastructure that lowers technical barriers to conducting pathway-specific analyses by enabling users without programming expertise to interact with complex biomedical data through natural language queries, thereby fostering inclusive, data-informed precision medicine.

## Figures and Tables

**Figure 1 ijms-26-06487-f001:**
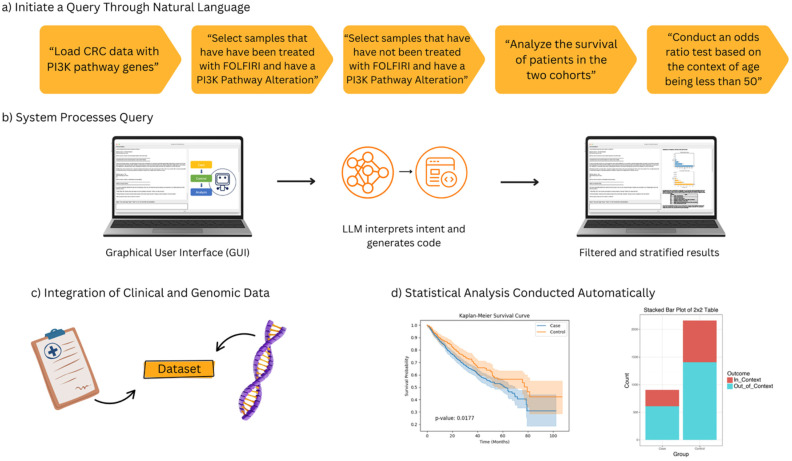
Overview of the AI-HOPE-PI3K workflow for pathway-specific precision oncology in colorectal cancer (CRC). This figure illustrates the end-to-end architecture and functionality of AI-HOPE-PI3K, a conversational artificial intelligence system designed for integrative analysis of *PI3K* pathway alterations in CRC. (**a**) Users initiate queries through natural language prompts, such as filtering by treatment exposure (e.g., FOLFIRI), mutation status in PI3K pathway genes (e.g., *PIK3CA*, *PTEN*), and age-related stratification (e.g., patients under 50 years). (**b**) The system processes the query through a graphical user interface (GUI) that leverages a large language model (LLM) to interpret semantic intent and translate it into executable code, producing filtered and stratified outputs. (**c**) Harmonized clinical and genomic data are integrated in real time, allowing the system to dynamically generate case–control cohorts for downstream analyses. (**d**) The automated statistical analysis capabilities of AI-HOPE-PI3K. On the left, a Kaplan–Meier survival plot shows a comparison of overall survival between two patient cohorts defined by *PI3K* pathway alteration and treatment exposure. The survival curves are automatically generated with log-rank test statistics and confidence intervals based on natural language queries. On the right, a bar plot presents the results of an odds ratio analysis comparing the frequency of a clinical or genomic feature—such as age group or mutation presence—across defined subgroups. These visualizations are produced in real time and serve to support rapid hypothesis generation and interpretation by end-users without the need for programming or manual plotting.

**Figure 2 ijms-26-06487-f002:**
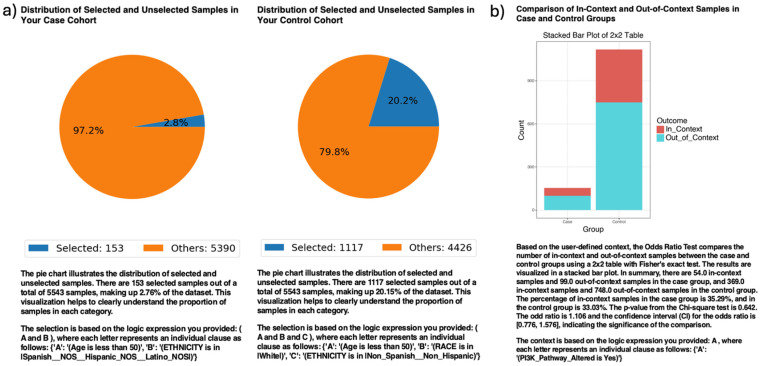
AI-HOPE-PI3K analysis of *PI3K* pathway alterations in early-onset colorectal cancer (EOCRC) among Hispanic/Latino (H/L) and Non-Hispanic White (NHW) patients. (**a**) Pie charts display the proportion of selected samples in each cohort after natural language-guided filtering of the harmonized dataset. The case cohort includes 153 EOCRC patients under age 50 identified as H/L, representing 2.76% of the dataset. The control cohort includes 1117 EOCRC patients under age 50 identified as Non-Hispanic White, representing 20.15% of the dataset. (**b**) A 2 × 2 odds ratio analysis evaluates the frequency of *PI3K* pathway alterations between the two groups. The stacked bar plot illustrates the distribution of samples with and without *PI3K* pathway alterations, labeled as “In_Context” and “Out_of_Context,” respectively. *PI3K* pathway alterations were present in 35.29% of H/L cases and 33.03% of NHW controls. The calculated odds ratio was 1.106 (95% CI: 0.776–1.576), with a *p*-value of 0.642, indicating no statistically significant difference.

**Figure 3 ijms-26-06487-f003:**
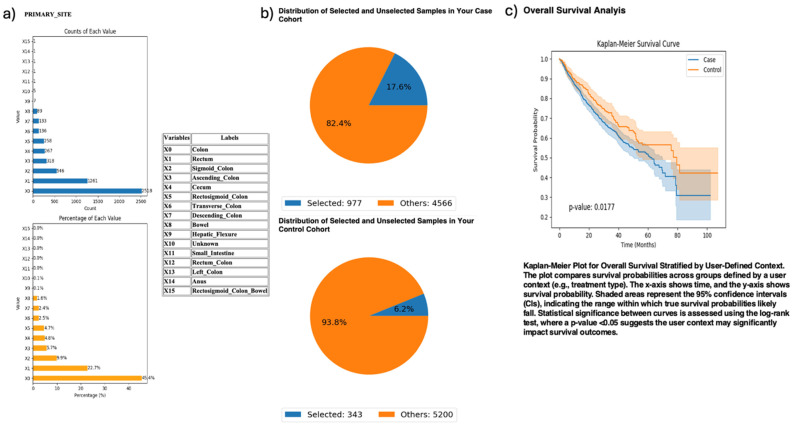
AI-HOPE-PI3K analysis of *PI3K*-altered colorectal cancer (CRC) samples by primary tumor location (colon vs. rectum). This figure illustrates the application of AI-HOPE-PI3K to evaluate survival outcomes among CRC patients with *PI3K* pathway alterations, stratified by primary tumor site. (**a**) Bar plots display the distribution of tumor site annotations across the dataset, with “Colon” (X0) and “Rectum” (X1) representing the most frequent anatomical sites. Colon samples constitute the majority of annotated entries (45.4%), while rectal tumors account for 22.7%. All other primary sites are shown for context. (**b**) Pie charts visualize the subset of samples selected through natural language-driven query filters. The case cohort includes 977 CRC patients with *PI3K*-altered tumors located in the colon (17.6% of the dataset), while the control cohort consists of 343 patients with *PI3K*-altered tumors in the rectum (6.2% of the dataset). (**c**) Kaplan–Meier survival curves compare overall survival between the two groups. CRC patients with *PI3K* pathway alterations originating in the colon exhibited significantly worse survival outcomes compared to those with rectal tumors (*p* = 0.0177). Shaded regions represent 95% confidence intervals.

**Figure 4 ijms-26-06487-f004:**
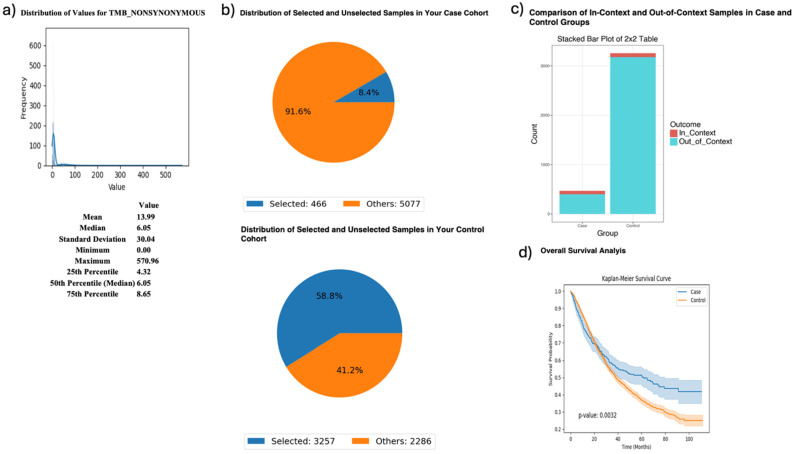
AI-HOPE-PI3K analysis of tumor mutational burden (TMB) and survival outcomes in colorectal cancer (CRC) patients treated with FOLFIRI chemotherapy, stratified by *MTOR* mutation status. This figure demonstrates the use of AI-HOPE-PI3K to evaluate survival and mutation context in CRC patients treated with the FOLFIRI regimen (Fluorouracil, Leucovorin, Irinotecan), stratified by TMB and analyzed for *MTOR* mutation enrichment. (**a**) A histogram displays the distribution of nonsynonymous TMB across all samples. The mean TMB is 13.99, with a median of 6.05 and a long right-tailed distribution. These metrics contextualize the threshold (TMB > 10) used to define the high-TMB case cohort. (**b**) Pie charts show the distribution of selected samples within each cohort following query-based stratification. The case cohort includes 466 high-TMB CRC patients treated with FOLFIRI (8.4% of the dataset), while the control cohort includes 3257 low-TMB CRC patients treated with the same regimen (58.8% of the dataset). (**c**) A 2 × 2 odds ratio analysis evaluates the enrichment of *MTOR* mutations between the two groups. The stacked bar plot shows that *MTOR* mutations were more frequently observed in the high-TMB group. However, detailed statistical outputs are not shown here. (**d**) Kaplan–Meier survival curves compare overall survival between high-TMB and low-TMB patients, both treated with FOLFIRI. Patients with high TMB exhibited significantly improved survival outcomes (*p* = 0.0032). Shaded areas represent 95% confidence intervals, indicating a robust survival benefit associated with elevated TMB in the context of FOLFIRI chemotherapy.

**Figure 5 ijms-26-06487-f005:**
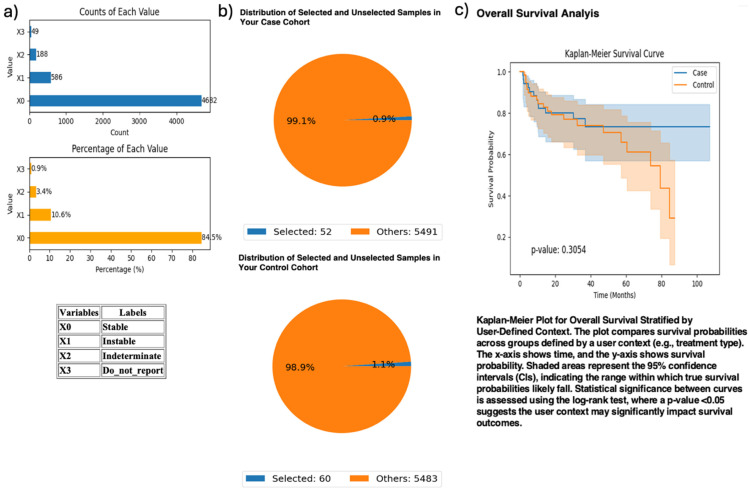
AI-HOPE-PI3K analysis of *PIK3CA* mutation status among microsatellite instability-high (MSI-H) colorectal cancer (CRC) patients treated with pembrolizumab. This figure demonstrates the use of AI-HOPE-PI3K to assess survival outcomes in MSI-H CRC patients receiving immunotherapy, stratified by *PIK3CA* mutation status. (**a**) Bar plots show the overall distribution of microsatellite instability types in the dataset. Samples classified as “Instable” (X1) account for 10.6% (*n* = 586), while “Stable” (X0) represents the majority (84.5%, *n* = 4682). Other MSI classifications, including “Indeterminate” and “Do_not_report,” appear less frequently. (**b**) Pie charts visualize cohort selection following natural language query execution. The case cohort includes 52 MSI-H CRC patients with *PIK3CA* mutations treated with pembrolizumab (0.9% of the dataset), while the control cohort includes 60 MSI-H patients without *PIK3CA* mutations treated with the same agent (1.1% of the dataset). (**c**) Kaplan–Meier survival analysis compares overall survival between the two groups. While *PIK3CA*-mutant MSI-H patients appear to have improved survival relative to wild-type counterparts, the difference was not statistically significant (*p* = 0.3054). Shaded regions represent the 95% confidence intervals, suggesting overlapping survival distributions and insufficient evidence of differential outcomes by *PIK3CA* status in this immunotherapy-treated subgroup.

## Data Availability

Data used in this study is available to the public and can be found at cbioportal.org. The AI-HOPE-PI3K software, along with demonstration data and documentation, is available at https://github.com/Velazquez-Villarreal-Lab/AI-PI3K (accessed on 15 May 2025).

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
