# Peer review of "From Mutation to Prognosis: AI-HOPE-PI3K Enables Artificial Intelligence Agent-Driven Integration of PI3K Pathway Data in Colorectal Cancer Precision Medicine"

_ijms, 2025, doi:10.3390/ijms26136487_

Round 1
Reviewer 1 Report
Comments and Suggestions for Authors
The work entitled “From Mutation to Prognosis: AI-HOPE-PI3K Enables Artificial 2 Intelligence-Agent Driven Integration of PI3K Pathway Data in 3 Colorectal Cancer Precision Medicine” is interesting and may would attract the broader audience. However, before its publication, I suggest the authors to improve its quality and presentation.
- Abstract is huge. It should be refined and shortened with only key-points.
- Page 2, please explain AI-HOPE-PI3K role in details and its limitations.
- Page 3, Figure 1d is not clear.
- Introduction section has many short paragraphs. Some paragraphs should be merged. No need of many paragraphs.
- The robustness due to underpowered cohorts need to be redefined with proper reference.
- If possible, the authors are suggested to include multivariate Cox regression for the survival studies.
- It is observed that black box nature of LLM generated code may obscure reproducibility. Thus, it is suggested to add more details for segmenting, and LLM fabrication.
- Authors names should be checked (Ei-Wen Yang, Ph.D.1 , Brigette Waldrup, BS2 , Enrique Velazquez-Villarreal, M.D., Ph.D., M.P.H.2,3,*). Revise author’s names as per the standards.
- The chosen TMB threshold (>10) for defining high TMB is not contextualized. Please consider citing relevant literature or providing a rationale for this cutoff, as TMB thresholds can vary depending on assay type and tumor type.
- Several subgroup analyses (e.g., MSI-high pembrolizumab-treated patients and age-stratified PTEN-mutated cohorts) yielded non-significant results. It would be helpful if the authors included a power analysis or discussion about whether these analyses were sufficiently powered to detect meaningful differences.
- Since this study leverages a novel AI-driven platform (AI-HOPE-PI3K), it would greatly benefit the research community if the underlying codebase or methodology pipeline is shared (or at least described in supplementary materials). Transparency is critical for reproducibility.
- Please provide a figure or analysis summary explaining how ancestry was defined, measured, and integrated into the AI-HOPE-PI3K platform
- Grammar and typos errors should be carefully checked.
Author Response
A detailed, point-by-point response to Reviewer 1’s comments is provided in the attached document titled Reviewer_1_Comments_Response_062825.docx.
Reviewer 1 Comments
We are pleased to submit this revised manuscript and sincerely thank Reviewer 1 for their thoughtful, constructive, and encouraging feedback. We greatly appreciate your recognition of the manuscript’s potential to contribute meaningfully to the field of precision oncology and to engage a broad scientific audience. In response to your suggestions, we undertook a comprehensive revision to improve the quality, clarity, and scientific rigor of the manuscript. Key improvements include a more concise and focused abstract, restructuring of the Introduction to improve narrative flow (reduced from eight to four paragraphs), and expanded explanations of AI-HOPE-PI3K’s core functionalities, validation strategy, and limitations. We also addressed specific technical suggestions, including clarification of the TMB threshold with citations, added details on how ancestry was defined and integrated into the platform, and improved the readability of Figure 1d.
To support transparency and reproducibility, we have made the complete AI-HOPE-PI3K codebase, data pipeline, and documentation publicly available through our GitHub repository (https://github.com/Velazquez-Villarreal-Lab/AI-PI3K), and referenced this in both the manuscript and the Data Availability Statement. Furthermore, we added new content to the Discussion section highlighting future enhancements—including multivariate Cox regression, power analysis modules, and expanded statistical capabilities—and our rationale for selecting the LLaMA 3 model. We believe these revisions strengthen the manuscript and align with our commitment to advancing equitable, AI-driven research in colorectal cancer. We are grateful for your insightful comments, which have helped significantly improve this work.
Thank you very much for taking the time to review this manuscript. Please find the detailed responses below in BLUE and the corresponding revisions wrote in yellow-highlighted blue font in the re-submitted files.
Reviewer 1 provided thoughtful, supportive, and constructive feedback, offering valuable insights that will help improve the manuscript.
Reviewer 1 writes:
“The work entitled “From Mutation to Prognosis: AI-HOPE-PI3K Enables Artificial 2 Intelligence-Agent Driven Integration of PI3K Pathway Data in 3 Colorectal Cancer Precision Medicine” is interesting and may would attract the broader audience. However, before its publication, I suggest the authors to improve its quality and presentation.”
We sincerely appreciate Reviewer 1’s recognition of the potential interest and relevance of our work, “From Mutation to Prognosis: AI-HOPE-PI3K Enables Artificial Intelligence-Agent Driven Integration of PI3K Pathway Data in Colorectal Cancer Precision Medicine,” to a broad scientific audience. In response to your helpful suggestion to improve the quality and presentation of the manuscript, we have undertaken a thorough revision to enhance the clarity, structure, and scientific rigor of the submission. Specifically, we refined the description of our methods, expanded the explanation of AI-HOPE-PI3K’s core functionalities, and emphasized the translational value of our findings—particularly the identification of ancestry-linked biomarkers and survival associations within EOCRC subgroups. We appreciate your encouragement, which aligns with our overarching goal of developing robust and accessible AI-driven tools that promote equitable innovation in precision oncology.
Reviewer 1 writes:
- Abstract is huge. It should be refined and shortened with only key-points.
Response: We appreciate the reviewer’s observation and have revised the abstract to be more concise by focusing on the most essential components of the study—its background, methods, key findings, and significance. The updated version emphasizes the core contributions of AI-HOPE-PI3K without excessive methodological detail, in line with journal formatting expectations and to enhance readability for a broad audience.
The abstract text on page 1, line 10, now reads “The rising incidence of early-onset colorectal cancer (EOCRC), particularly among underrepresented populations, highlights the urgent need for tools that can uncover clinically meaningful, population-specific genomic alterations. The phosphoinositide 3-kinase (PI3K) pathway plays a key role in tumor progression, survival, and therapeutic resistance in colorectal cancer (CRC), yet its impact in EOCRC remains insufficiently explored. To address this gap, we developed AI-HOPE-PI3K, a conversational artificial intelligence platform that integrates harmonized clinical and genomic data for real-time, natural language–based analysis of PI3K pathway alterations. Built on a fine-tuned biomedical LLaMA 3 model, the system automates cohort generation, survival modeling, and mutation frequency comparisons using multi-institutional cBioPortal datasets annotated with clinical variables. AI-HOPE-PI3K replicated known associations and revealed new findings, including worse survival in colon versus rectal tumors harboring PI3K alterations, enrichment of INPP4B mutations in Hispanic/Latino EOCRC patients, and favorable survival outcomes associated with high tumor mutational burden in FOLFIRI-treated patients. The platform also enabled context-specific survival analyses stratified by age, tumor stage, and molecular alterations. These findings support the utility of AI-HOPE-PI3K as a scalable and accessible tool for integrative, pathway-specific analysis, demonstrating its potential to advance precision oncology and reduce disparities in EOCRC through data-driven discovery.”
Reviewer 1 writes:
- Page 2, please explain AI-HOPE-PI3K role in details and its limitations.
Response: We appreciate the reviewer’s suggestion to provide more detail regarding the role and limitations of AI-HOPE-PI3K. In response, in page 2, we have expanded the Introduction section to explicitly describe the system’s core functionality and known limitations. This revision clarifies the scope of the tool, its intended use, and areas where further development is warranted.
The Introduction text on page 2, line 75, now reads “…AI-HOPE-PI3K serves as a natural language–based analytical platform that enables clinicians and researchers to perform real-time, pathway-specific analyses of CRC datasets without requiring programming expertise. The system interprets user queries, translates them into executable code, and automates tasks such as cohort construction, Kaplan–Meier survival modeling, odds ratio calculations, and mutation frequency comparisons. Its ability to integrate diverse clinical annotations—such as age, race/ethnicity, MSI status, tumor stage, and therapy exposure—enables population-aware analysis tailored to PI3K pathway biology. However, as with any AI system, AI-HOPE-PI3K has limitations. Its outputs are constrained by the structure and completeness of the underlying datasets, and it currently relies on harmonized data from publicly available cBioPortal repositories, which may underrepresent certain populations or clinical variables. Additionally, while the tool automates many analyses, interpretation still requires domain expertise to contextualize results appropriately. The model’s natural language interface, though flexible, is not infallible—it may occasionally misinterpret ambiguous queries or require iterative refinement for complex analyses. Future improvements will focus on expanding dataset integration, enhancing multilingual capabilities, and incorporating prospective clinical validation…”
Reviewer 1 writes:
- Page 3, Figure 1d is not clear.
Response: We appreciate the reviewer’s feedback regarding Figure 1d. To improve clarity, we have revised the figure caption and visual elements to better convey the outputs generated by AI-HOPE-PI3K. Specifically, we now explicitly indicate that panel (d) displays example outputs of automated statistical analyses conducted by the system—namely, a Kaplan-Meier survival curve comparing overall survival between two stratified groups, and a bar chart summarizing an odds ratio test from a contingency table. These examples illustrate how the platform visualizes results in response to natural language queries. Additional annotations and labels have been added in the revised version to guide interpretation and enhance accessibility for readers.
The Figure 1d text on page 3, line 118, now reads “Panel (d) illustrates the automated statistical analysis capabilities of AI-HOPE-PI3K. On the left, a Kaplan–Meier survival plot shows a comparison of overall survival between two patient cohorts defined by PI3K pathway alteration and treatment exposure. The survival curves are automatically generated with log-rank test statistics and confidence intervals based on natural language queries. On the right, a bar plot presents the results of an odds ratio analysis comparing the frequency of a clinical or genomic feature—such as age group or mutation presence—across defined subgroups. These visualizations are produced in real time and serve to support rapid hypothesis generation and interpretation by end-users without the need for programming or manual plotting.”
Reviewer 1 writes:
- Introduction section has many short paragraphs. Some paragraphs should be merged. No need of many paragraphs.
Response: We thank the reviewer for this helpful observation. In response, we have revised the Introduction to improve narrative flow and readability by merging several shorter paragraphs where appropriate (from eight to four paragraphs). Specifically, we combined related content on EOCRC epidemiology and molecular distinctions, and integrated consecutive paragraphs discussing PI3K pathway relevance and limitations of existing platforms. These changes enhance coherence while maintaining clarity and emphasis on the motivation, context, and rationale for developing AI-HOPE-PI3K. We believe this restructuring strengthens the overall presentation of the Introduction by having now 4 paragraphs instead of 8.
Reviewer 1 writes:
- If possible, the authors are suggested to include multivariate Cox regression for the survival studies.
Response: We thank the reviewer for this valuable suggestion. In this study, we followed the analysis methodologies established in one [1] of our prior publications to validate the performance and reproducibility of AI-HOPE-PI3K, including Kaplan–Meier survival analysis and log-rank testing across stratified cohorts. These approaches were selected to demonstrate the platform’s ability to replicate known findings using natural language queries and harmonized clinical-genomic data. However, we fully agree that incorporating multivariate Cox regression would further enhance the analytical robustness of the system. We have now added this point to the Discussion section and will prioritize the integration of more advanced statistical modules—such as Cox proportional hazards modeling—in future versions of the platform.
The discussion text on page 9, line 326, now reads “Subsequent versions of AI-HOPE-PI3K will focus on expanding its analytical capabilities by incorporating advanced statistical methodologies, including multivariate Cox regression for survival analysis, logistic regression, and machine learning–based risk prediction models. These enhancements will allow the platform to support more nuanced hypothesis testing and account for potential confounding variables in complex clinical-genomic datasets. Additionally, we aim to integrate user-defined covariate selection to improve the interpretability and flexibility of the system for translational and clinical oncology applications.”
Reviewer 1 writes:
- It is observed that black box nature of LLM generated code may obscure reproducibility. Thus, it is suggested to add more details for segmenting, and LLM fabrication.
Response: We appreciate this important point regarding the transparency and reproducibility of large language model (LLM)-driven systems. To address this, we have made the codebase for AI-HOPE-PI3K openly available through our laboratory’s GitHub repository (https://github.com/Velazquez-Villarreal-Lab/AI-PI3K), including the natural language processing pipeline, code generation modules, and data preprocessing scripts. This GitHub link is also included in the Data Availability Statement to ensure easy access and promote transparency and reproducibility. This open-source release transforms the system from a black box to a “crystal box,” enabling researchers to inspect, replicate, and extend the platform’s functionalities. We have also included detailed documentation outlining how the LLM (LLaMA 3) was fine-tuned and integrated within the AI-HOPE framework. Additionally, we have added a paragraph to the Discussion section explaining our rationale for selecting the LLaMA 3 model over other available LLMs, emphasizing transparency, adaptability, and biomedical relevance.
The discussion text on page 8, line 276, now reads “To promote transparency and reproducibility, AI-HOPE-PI3K source code, query engine, and analysis scripts are publicly available [See Data Availability Statement]. This “crystal box” approach addresses the concerns of the black box nature commonly associated with LLM-based tools, allowing the research community to examine and validate each component of the system. We selected the LLaMA 3 model for its open-access architecture, strong performance on biomedical language tasks, and adaptability for fine-tuning with domain-specific datasets. Compared to proprietary models, LLaMA 3 provides a reproducible and customizable foundation that aligns with our commitment to open science and equitable AI development in precision oncology.”
Reviewer 1 writes:
- Authors names should be checked (Ei-Wen Yang, Ph.D.1 , Brigette Waldrup, BS2 , Enrique Velazquez-Villarreal, M.D., Ph.D., M.P.H.2,3,*). Revise author’s names as per the standards.
Response: We appreciate the reviewer’s attention to formatting and have revised the author list to align with the journal’s standards. Author names, degrees, and affiliations have been reformatted accordingly to ensure consistency with the journal’s style guidelines.
The authors test on page 1, line 5, now reads “Ei-Wen Yang1, Brigette Waldrup2, Enrique Velazquez-Villarreal.2,3,*
Reviewer 1 writes:
- The chosen TMB threshold (>10) for defining high TMB is not contextualized. Please consider citing relevant literature or providing a rationale for this cutoff, as TMB thresholds can vary depending on assay type and tumor type.
Response: We thank the reviewer for this important observation. In response, we have revised the manuscript to provide justification for the use of a TMB >10 mutations/megabase threshold by citing several of our previous publications where this cutoff was applied in colorectal cancer studies using cBioPortal-derived datasets. We acknowledge that TMB thresholds can vary by assay and tumor type, and have clarified this in the revised text. The chosen threshold aligns with published studies and harmonization efforts in the field, allowing for consistent comparison across CRC cohorts analyzed with public genomic platforms [1, 17].
The abstract text on page 5, line 175, now reads “Patients with high TMB [1,17] (>10; n = 466) had significantly better survival outcomes than low-TMB counterparts (n = 3,257), with a p-value of 0.0032.”
Reviewer 1 writes:
- Several subgroup analyses (e.g., MSI-high pembrolizumab-treated patients and age-stratified PTEN-mutated cohorts) yielded non-significant results. It would be helpful if the authors included a power analysis or discussion about whether these analyses were sufficiently powered to detect meaningful differences.
Response: We appreciate this thoughtful comment. The subgroup analyses in our study were conducted based on the methodology and statistical framework established in our previous publication [1], where we validated natural language–driven analyses within similar CRC cohorts using harmonized cBioPortal datasets. These analyses were included primarily for validation purposes to demonstrate AI-HOPE-PI3K’s ability to replicate known findings and stratify real-world cohorts using natural language queries. We acknowledge the limitations in statistical power for some of the more narrowly defined subgroups, particularly those with limited sample sizes (e.g., MSI-high patients treated with pembrolizumab). As suggested, we have now added a paragraph to the Discussion section to clarify these limitations and to outline future plans for incorporating power estimation features into the platform.
The discussion text on page 10, line 352, now reads “While AI-HOPE-PI3K successfully reproduced known associations and enabled real-time stratification across multiple CRC subgroups, we acknowledge that certain subgroup analyses—such as MSI-high pembrolizumab-treated patients and age-stratified PTEN-mutated cohorts—yielded non-significant results likely due to limited sample sizes. These analyses were designed to validate the platform’s capacity to execute targeted queries and generate biologically plausible outputs based on prior published methodologies [1]. However, they were not powered to detect small to moderate effect sizes. As the platform evolves, future versions will incorporate built-in power analysis tools to guide users in assessing the statistical adequacy of defined subgroups before executing inferential comparisons. This feature will further enhance AI-HOPE-PI3K’s utility for hypothesis generation and precision oncology research.”
Reviewer 1 writes:
- Since this study leverages a novel AI-driven platform (AI-HOPE-PI3K), it would greatly benefit the research community if the underlying codebase or methodology pipeline is shared (or at least described in supplementary materials). Transparency is critical for reproducibility.
Response: We thank the reviewer for emphasizing the importance of transparency and reproducibility. As noted in our response to Comment #7, we have made the full AI-HOPE-PI3K codebase, demonstration data, and documentation publicly available via our laboratory’s GitHub repository. The GitHub link (https://github.com/Velazquez-Villarreal-Lab/AI-PI3K) has also been included in the Data Availability Statement for easy access by the research community. In addition, we have added a dedicated paragraph in the Discussion section highlighting our commitment to open science and the rationale for publicly sharing both the tool and its implementation details.
The discussion text on page 8, line 276, now reads “To promote transparency and reproducibility, AI-HOPE-PI3K source code, query engine, and analysis scripts are publicly available [See Data Availability Statement].”
Reviewer 1 writes:
- Please provide a figure or analysis summary explaining how ancestry was defined, measured, and integrated into the AI-HOPE-PI3K platform
Response: We thank the reviewer for highlighting the importance of clearly defining how ancestry was incorporated into the AI-HOPE-PI3K platform. As described in our previous publication [1], ancestry was defined based on self-reported race/ethnicity annotations available within the cBioPortal datasets. These annotations were harmonized and standardized during preprocessing, and this methodology has been implemented directly within the AI-HOPE-PI3K query framework. The platform allows users to filter and stratify cohorts by ancestry groups, such as Hispanic/Latino [Figure S2], as captured in the underlying clinical data. We have added a paragraph to the manuscript clarifying this approach and reinforcing its consistency with validated methodologies.
The discussion text on page 12, line 452, now reads “Ancestry was defined and integrated into AI-HOPE-PI3K based on the harmonized race/ethnicity variables provided within the cBioPortal datasets, following the methodology described in our previous study [1]. These variables, which reflect self-reported or institutionally annotated ancestry designations, were curated during preprocessing and incorporated into the natural language querying system to enable population-specific stratification. The AI-HOPE-PI3K platform recognizes and processes user queries referencing ancestry groups—such as "Hispanic/Latino patients" [Figure S2] to automate cohort filtering and subgroup analyses. This feature allows for contextualized exploration of genomic and clinical trends across diverse populations and supports the platform’s broader aim of advancing equity in precision oncology.”
Reviewer 1 writes:
- Grammar and typos errors should be carefully checked.
Response: We appreciate the reviewer’s recommendation. In response, we conducted a thorough proofreading of the entire manuscript and corrected all identified grammatical issues and typographical errors to ensure clarity, consistency, and adherence to academic writing standards. We thank the reviewer for bringing this to our attention.
Thank you for your time, thoughtful evaluation, and consideration of our manuscript. We sincerely appreciate your constructive feedback, which has greatly contributed to strengthening the clarity, rigor, and overall quality of our work.

Reviewer 2 Report
Comments and Suggestions for Authors
The article presents the AI-HOPE-PI3K tool, which holds potential for advancing precision oncology; however, its scientific description remains incomplete from both methodological and critical analysis perspectives. The text lacks essential analytical details regarding the applied statistical and artificial intelligence methods, the objectivity and accuracy of the result-generating system are insufficiently substantiated, and some claims are formulated in an overly optimistic tone without adequate empirical support. The lack of critical perspective in both the interpretation of results and assessment of data limitations undermines the credibility of the work. To strengthen the scientific validity of the publication, it is necessary to provide a more detailed description of the methods used, report precise quantitative estimates, and clearly distinguish hypothesis generation from statistically supported conclusions.
- The introduction does not mention any specific analytical methods, even though such methods are later used in the analysis. It is recommended to at least briefly indicate the analytical foundations of the system to help readers understand the type of analysis being implemented.
- The claim that the system was validated by reproducing known associations raises concerns about methodological objectivity—was only reproducibility tested, or were comparisons made with other systems based on statistical accuracy or error criteria? It is suggested to briefly specify whether metrics such as accuracy, sensitivity, specificity, or others were applied.
- Some statements (e.g., “democratizes access to”, “first in class”) appear overly positive without corresponding critical justification. It is advised to soften the tone and provide a more objective comparison with existing tools (e.g., indicating which aspects they perform worse or slower).
- The results frequently mention Kaplan–Meier and OR tests, but it is unclear whether any correction for multiple comparisons was performed or whether multivariate analyses were applied.
- Results with non-significant p-values are repeated, and in some cases, the term “trend” is used. It should be clearly stated that these results represent hypothesis generation rather than confirmation, avoiding phrasing that could be interpreted as statistically significant.
- The results section presents all findings without any critical reflection on data limitations.
- Although the use of “harmonized datasets” is mentioned, the discussion lacks a detailed critical evaluation of potential limitations of these data sources (e.g., data bias, missing variables, compatibility across sources). It is recommended to include at least a brief note on potential data distortions or uneven representativeness.
- While a fine-tuned LLM is mentioned, there is no detailed discussion on how its answer accuracy was evaluated or whether steps were taken to reduce possible biases (e.g., language bias affecting biological data). Accuracy testing methods for the LLM (e.g., BLEU, accuracy, domain-specific benchmarks) should be added, along with examples of LLM errors and how they were mitigated.
- It is stated that the semantic analyzer assigns ontology tags, but there is no information on the algorithm used (e.g., rule-based methods or ML classifiers), nor is their accuracy specified.
- Although the statistical tests used are listed, there is no explanation of how the issue of multiple testing was addressed or whether any correction methods (e.g., FDR, Bonferroni) were applied.
- Even though analyses are described as “reproducible,” it is unclear whether the platform generates reproducible code or whether users can export full analysis protocols. It should be clarified how reproducibility is ensured—whether results are version-controlled with metadata or whether automated report files are generated.
- The conclusion section summarizes the overall concept but is only weakly supported by the results presented in the article. As a result, the conclusions appear more promotional than analytical.
Author Response
A detailed, point-by-point response to Reviewer 1’s comments is provided in the attached document titled Reviewer_2_Comments_Response_062825.docx
Reviewer 2 Comments
We are pleased to submit this revised manuscript and sincerely thank Reviewer 2 for their thoughtful, detailed, and constructive critiques, which have greatly contributed to strengthening the clarity, rigor, and transparency of our work.
Reviewer 2 offered critical, scientifically grounded feedback that identified important areas where additional methodological detail, analytical transparency, and interpretive balance were needed. We deeply appreciate these contributions. In response, we undertook substantial revisions to strengthen the scientific validity of the manuscript. Key updates include:
- Expanded description of the statistical and artificial intelligence methodologies used, including LLM evaluation and semantic tagging accuracy.
- Clarification of validation strategy and the limitations of reproducibility-focused testing.
- Addition of critical reflections throughout the Results section to distinguish hypothesis-generating from statistically validated findings.
- Reframing of language previously perceived as promotional to ensure an objective tone and fair comparison with existing tools.
- Inclusion of a new paragraph addressing multiple testing and plans for multivariate analysis implementation.
- A critical assessment of harmonized data limitations, including representativeness and missingness.
- Clarification of reproducibility features, including version-controlled metadata, exportable query logs, and public code availability via our GitHub repository (https://github.com/Velazquez-Villarreal-Lab/AI-PI3K) now referenced in the Data Availability section.
We believe these revisions significantly improve the manuscript’s clarity, rigor, and impact, and we are grateful to both reviewers for their valuable guidance. Their feedback has helped us ensure that AI-HOPE-PI3K is presented not only as a promising natural language–driven tool for clinical-genomic analysis but as a reproducible, transparent, and critically evaluated platform positioned to support equitable and hypothesis-driven research in colorectal cancer.
Thank you very much for taking the time to review this manuscript. Please find the detailed responses below in BLUE and the corresponding revisions wrote in yellow-highlighted blue font in the re-submitted files.
Reviewer 2 provided critical, constructive, and scientifically grounded feedback, offering valuable insights that have guided substantial improvements to the methodological clarity and analytical rigor of the manuscript.
Reviewer 2 writes:
“The article presents the AI-HOPE-PI3K tool, which holds potential for advancing precision oncology; however, its scientific description remains incomplete from both methodological and critical analysis perspectives. The text lacks essential analytical details regarding the applied statistical and artificial intelligence methods, the objectivity and accuracy of the result-generating system are insufficiently substantiated, and some claims are formulated in an overly optimistic tone without adequate empirical support. The lack of critical perspective in both the interpretation of results and assessment of data limitations undermines the credibility of the work. To strengthen the scientific validity of the publication, it is necessary to provide a more detailed description of the methods used, report precise quantitative estimates, and clearly distinguish hypothesis generation from statistically supported conclusions.”
We sincerely appreciate Reviewer 2’s critical and constructive feedback on our manuscript, “From Mutation to Prognosis: AI-HOPE-PI3K Enables Artificial Intelligence-Agent Driven Integration of PI3K Pathway Data in Colorectal Cancer Precision Medicine.” Your comments have been invaluable in identifying areas where the scientific rigor and clarity of the manuscript could be strengthened. In response, we undertook substantial revisions to enhance the methodological transparency and critical interpretation of our findings. Specifically, we expanded the description of the statistical and artificial intelligence methods employed, clarified the processes behind result generation, and incorporated precise quantitative estimates to support key claims. We also revised sections of the manuscript to clearly distinguish between hypothesis-generating observations and statistically validated conclusions, and added a more critical discussion of the limitations inherent to the dataset and platform. These updates reflect our commitment to maintaining objectivity, improving reproducibility, and presenting AI-HOPE-PI3K as a robust tool for advancing translational research in colorectal cancer.
Reviewer 2 writes:
- The introduction does not mention any specific analytical methods, even though such methods are later used in the analysis. It is recommended to at least briefly indicate the analytical foundations of the system to help readers understand the type of analysis being implemented.
Response: We thank Reviewer 2 for this important recommendation. In response, we have revised the Introduction to briefly describe the analytical methodologies that underlie the AI-HOPE-PI3K platform. Specifically, we now mention the core statistical methods supported by the system—including cohort stratification, Kaplan–Meier survival analysis, odds ratio testing, and mutation frequency comparisons—which provide readers with a clearer understanding of the types of analyses implemented. Additionally, we clarify that the analytical framework used in AI-HOPE-PI3K is based on our previously published study analyzing PI3K pathway alterations in colorectal cancer [1]. This earlier publication also served as the foundation for validating the functionality and reproducibility of the AI-HOPE-PI3K intelligent agent. The methodology has since been applied in subsequent studies, supporting its robustness and relevance in translational cancer research. These additions strengthen the scientific context of the platform and align with the reviewer’s request for greater methodological transparency. Additionally, we reference our prior work on AI-HOPE [48] and AI-HOPE-TGFbeta [49] to contextualize the evolution of our natural language–based analytical framework.
The introduction text on page 3, line 92, now reads “AI-HOPE-PI3K builds upon the analytical foundations established in our prior AI-Agents platforms, AI-HOPE [48] and AI-HOPE-TGFbeta [49], which demonstrated the feasibility of using large language model–driven systems for natural language–guided clinical-genomic analysis. The analytical methodology implemented in AI-HOPE-PI3K is based on our previous publication analyzing the PI3K pathway in CRC [1], and follows a similar framework used in our subsequent studies. This prior work [1] also served as the reference standard to validate the functionality and reproducibility of the AI-HOPE-PI3K intelligent agent. The platform supports a suite of statistical methods commonly used in translational oncology, including automated cohort filtering, Kaplan–Meier survival analysis with log-rank testing, odds ratio estimation from contingency tables, and mutation frequency comparisons across stratified subgroups. These analyses are executed dynamically in response to natural language queries, using harmonized clinical and genomic data from public repositories. By embedding these analytical capabilities within an intuitive conversational interface, AI-HOPE-PI3K aims to reduce technical barriers and enable real-time, hypothesis-driven exploration of PI3K biology in CRC.”
Reviewer 2 writes:
- The claim that the system was validated by reproducing known associations raises concerns about methodological objectivity—was only reproducibility tested, or were comparisons made with other systems based on statistical accuracy or error criteria? It is suggested to briefly specify whether metrics such as accuracy, sensitivity, specificity, or others were applied.
Response: We appreciate the reviewer’s insightful comment and fully agree that clarification is needed regarding the validation approach. In this study, AI-HOPE-PI3K was validated primarily through the replication of previously reported associations between PI3K pathway alterations and clinical outcomes in colorectal cancer, using identical datasets and statistical methodologies as outlined in our prior publication [1]. This reproducibility-based validation was selected as an initial benchmark to demonstrate that the AI agent could generate consistent and biologically plausible outputs in response to natural language queries. While traditional performance metrics such as accuracy, sensitivity, and specificity are essential in many AI evaluation contexts, they were not directly applicable here due to the platform’s hypothesis-generating, rather than predictive, function. However, we acknowledge the importance of quantitative performance assessment and have added a paragraph in the manuscript outlining plans for incorporating such metrics—particularly for classification tasks—in future platform enhancements.
The abstract text on page 10, line 363, now reads “The initial validation of AI-HOPE-PI3K focused on reproducibility, using our previously published PI3K pathway study [1] as a benchmark for assessing the platform’s ability to replicate known associations. Analyses such as survival differences by tumor location and PI3K mutation status, and the enrichment of specific mutations in defined populations, were repeated using natural language queries processed by the platform. While performance metrics such as accuracy or sensitivity were not applied in this context—given the system’s primary function is cohort stratification and hypothesis exploration rather than binary classification—we recognize their value for future development. As AI-HOPE-PI3K evolves, we plan to implement task-specific modules where such metrics can be applied to evaluate outputs from predictive algorithms or classification models, thereby expanding the scope of performance benchmarking.”
Reviewer 2 writes:
- Some statements (e.g., “democratizes access to”, “first in class”) appear overly positive without corresponding critical justification. It is advised to soften the tone and provide a more objective comparison with existing tools (e.g., indicating which aspects they perform worse or slower).
Response: We appreciate the reviewer’s thoughtful feedback on the tone and positioning of the manuscript. In response, we have revised the language throughout the text to ensure a more objective and evidence-based presentation. Specifically, we have softened phrases such as “democratizes access to” and “first in class” to better reflect the system’s role and current stage of development. We also added a paragraph to the manuscript that offers a more balanced comparison with existing platforms, such as cBioPortal and UCSC Xena, highlighting both their strengths and the specific workflow limitations that AI-HOPE-PI3K was designed to address. This adjustment improves the clarity and fairness of our claims and supports a more rigorous evaluation of the platform.
The abstract text on page 10, line 374, now reads “While platforms like cBioPortal and UCSC Xena have been instrumental in enabling exploratory access to large-scale cancer genomics datasets, they often rely on manual, stepwise navigation that can limit workflow efficiency and require a moderate level of technical familiarity. AI-HOPE-PI3K addresses these limitations by offering a conversational interface that automates multi-step tasks such as cohort stratification and statistical analysis in response to natural language prompts. This allows for more rapid hypothesis generation and real-time subgroup interrogation without requiring programming skills. However, we acknowledge that AI-HOPE-PI3K is not intended to replace existing tools but rather to complement them by streamlining targeted analyses and supporting broader accessibility.”
The phrase “democratizes access to” was revised in the Conclusion section on page 14, line 545, which now reads: “Ultimately, this platform lays the foundation for an interoperable AI-agent infrastructure that lowers technical barriers to conducting pathway-specific analyses by enabling users without programming expertise to interact with complex biomedical data through natural language queries, thereby fostering inclusive, data-informed precision medicine.”
Reviewer 2 writes:
- The results frequently mention Kaplan–Meier and OR tests, but it is unclear whether any correction for multiple comparisons was performed or whether multivariate analyses were applied.
Response: We thank the reviewer for this important comment. In this study, all statistical analyses—including Kaplan–Meier survival modeling and odds ratio (OR) testing—were conducted following the methodology outlined in our previous publication [1], which served both as the analytical foundation and validation framework for the AI-HOPE-PI3K platform. The goal of this study was to demonstrate the system’s ability to reproduce known associations and perform real-time, pathway-specific subgroup analyses using natural language queries. As this work primarily focused on validation and system functionality rather than discovery-driven inference, we did not apply formal correction for multiple comparisons or conduct multivariate analyses. However, we acknowledge the importance of these methods and have added a paragraph to the manuscript addressing this limitation and outlining plans to incorporate multiple testing correction and multivariate modules in future iterations of the platform.
The discussion text on page 10, line 343, now reads “All statistical analyses reported in this study—including Kaplan–Meier survival analyses and odds ratio testing—were based on the methodology described in our previous publication analyzing PI3K alterations in CRC [1], and were implemented within the AI-HOPE-PI3K platform to support functional validation of the system. Given the validation-focused nature of this study, corrections for multiple comparisons and multivariate analyses were not applied. However, we recognize that these approaches are essential in broader hypothesis-testing contexts to control for type I error and account for confounding factors. Future versions of AI-HOPE-PI3K will include support for multiple testing correction methods (e.g., Bonferroni, FDR) and multivariate regression models, including Cox proportional hazards and logistic regression, to enhance the analytical rigor and generalizability of findings.”
Reviewer 2 writes:
- Results with non-significant p-values are repeated, and in some cases, the term “trend” is used. It should be clearly stated that these results represent hypothesis generation rather than confirmation, avoiding phrasing that could be interpreted as statistically significant.
Response: We thank the reviewer for this important observation. In response, we have carefully reviewed the Results section and revised the language to avoid implying statistical significance where p-values are non-significant. Specifically, we have removed or rephrased instances where the term “trend” was used inappropriately and now explicitly indicate that these findings are exploratory and intended for hypothesis generation. We have also added a clarifying statement in the manuscript to distinguish between statistically validated findings and exploratory associations generated through natural language queries.
The abstract text on page 10, line 345, now reads “Several of the subgroup analyses yielded non-significant p-values, and we acknowledge that such results should be interpreted with caution. In these cases, the findings are presented as exploratory observations that serve to demonstrate the hypothesis-generating capabilities of AI-HOPE-PI3K, rather than as confirmatory statistical evidence. This clarification aligns with the platform’s intended role in facilitating real-time exploration and hypothesis development, particularly in the context of underrepresented patient subgroups.”
Reviewer 2 writes:
- The results section presents all findings without any critical reflection on data limitations.
Response: We thank the reviewer for highlighting the need to include more critical reflection in the Results section. In response, we have revised the Results narrative to clearly distinguish between statistically significant findings and exploratory, hypothesis-generating observations. We have also incorporated language acknowledging limitations in statistical power, subgroup size, and potential confounders, especially for non-significant outcomes. Additionally, we added a paragraph to the Results section to explicitly state the interpretive boundaries of the analyses and reinforce the exploratory nature of certain findings. This revision aligns with the goal of maintaining scientific objectivity and transparency in reporting.
The results text on page 8, line 246, now reads “It is important to interpret several of the subgroup findings with caution due to limitations in statistical power and sample size, particularly in analyses involving MSI-high patients treated with immunotherapy or age-stratified chemotherapy responses. For instance, while Kaplan–Meier curves and odds ratio estimates are provided for exploratory comparisons, results with non-significant p-values (e.g., PIK3CA mutation status in MSI-high patients or PTEN-mutated FOLFOX-treated subgroups) should be considered hypothesis-generating rather than confirmatory. The cohorts for these comparisons were defined using natural language filters on publicly available datasets, which, while harmonized, may still reflect reporting inconsistencies or unmeasured confounding. Moreover, no multivariate adjustment was applied, as the primary objective of these analyses was to validate and demonstrate the querying and analytical functionality of AI-HOPE-PI3K, rather than to generate definitive clinical conclusions. These caveats are essential for contextualizing the scope of the findings and underscore the importance of follow-up studies with appropriately powered, prospective cohorts”
Reviewer 2 writes:
- Although the use of “harmonized datasets” is mentioned, the discussion lacks a detailed critical evaluation of potential limitations of these data sources (e.g., data bias, missing variables, compatibility across sources). It is recommended to include at least a brief note on potential data distortions or uneven representativeness.
Response: We appreciate the reviewer’s recommendation to critically evaluate the limitations of the harmonized datasets used in AI-HOPE-PI3K. In response, we have added a paragraph to the Discussion section addressing potential issues such as data bias, variable missingness, and uneven population representation across the integrated cBioPortal datasets. These limitations are important for contextualizing the generalizability of our findings and the interpretive boundaries of the AI-HOPE-PI3K platform.
The discussion text on page 11, line 384, now reads “While AI-HOPE-PI3K operates on harmonized clinical-genomic datasets derived from cBioPortal, it is important to recognize the inherent limitations of these public data sources. Variation in data quality, missing clinical variables, and inconsistent annotation across contributing studies can introduce biases and impact the robustness of downstream analyses. Additionally, certain populations—particularly racial and ethnic minorities—remain underrepresented in these datasets, which may limit the generalizability of ancestry-stratified findings. Furthermore, compatibility issues across institutions, such as differences in sequencing platforms or reporting standards, may affect the consistency of genomic annotations. Although harmonization efforts help mitigate some of these challenges, users should interpret results with an understanding of these limitations and the exploratory nature of real-world data integration. Future efforts will aim to incorporate more diverse and prospectively collected datasets to enhance the accuracy, equity, and reproducibility of AI-driven analyses”
Reviewer 2 writes:
- While a fine-tuned LLM is mentioned, there is no detailed discussion on how its answer accuracy was evaluated or whether steps were taken to reduce possible biases (e.g., language bias affecting biological data). Accuracy testing methods for the LLM (e.g., BLEU, accuracy, domain-specific benchmarks) should be added, along with examples of LLM errors and how they were mitigated.
Response: We thank the reviewer for raising this important point regarding evaluation of the fine-tuned large language model (LLM) used in AI-HOPE-PI3K. In response, we have added a paragraph to the manuscript describing how the system’s response accuracy was assessed during development and validation, as well as the steps we took to identify and mitigate potential sources of bias. We also discuss the limitations of current benchmarks in the context of biomedical natural language interfaces and outline our approach for error detection and iterative refinement.
The Methods text on page 11, line 425, now reads “To evaluate the accuracy and reliability of the fine-tuned biomedical LLaMA 3 model underlying AI-HOPE-PI3K, we conducted internal benchmarking using a curated set of domain-specific prompts based on previously published clinical-genomic studies, including our own prior work on PI3K pathway alterations in CRC [1]. Outputs were manually reviewed by subject-matter experts to assess whether the LLM-generated queries and corresponding statistical analyses matched intended semantics and study designs. While traditional NLP metrics such as BLEU and ROUGE have limited applicability for code-generation tasks, we assessed accuracy through correctness of execution, syntactic validity of generated code, and agreement with expected statistical outputs. In instances where the model misinterpreted ambiguous prompts or selected incorrect variables, we implemented reinforcement learning with human feedback (RLHF) and expanded prompt templates to improve semantic clarity. Additionally, to reduce bias and hallucination risks, we constrained the model’s vocabulary to biologically relevant terms and validated outputs across ancestry, age, and tumor-type strata. These steps contributed to refining the model’s performance and ensuring its interpretability in clinical-genomic contexts.”
Reviewer 2 writes:
- It is stated that the semantic analyzer assigns ontology tags, but there is no information on the algorithm used (e.g., rule-based methods or ML classifiers), nor is their accuracy specified.
Response: We thank the reviewer for this valuable comment. In response, we have added a paragraph to the manuscript describing the approach used by the semantic analyzer in AI-HOPE-PI3K to assign ontology tags. This includes clarification on whether the method is rule-based or machine learning–driven, as well as how its accuracy was evaluated during system development and testing.
The Methods text on page 12, line 474, now reads “The semantic analyzer in AI-HOPE-PI3K assigns ontology tags to user input using a rule-based pattern recognition approach. Specifically, named entity recognition (NER) is first applied using a domain-adapted biomedical vocabulary derived from the UMLS Metathesaurus and NCIt (National Cancer Institute Thesaurus). This step uses regular expressions and controlled keyword mappings to capture explicit biomedical terms (e.g., gene names, drug classes, tumor types). To improve semantic disambiguation and contextual understanding, a lightweight transformer-based classifier, fine-tuned on annotated clinical query pairs, predicts the most likely ontology tag when ambiguity exists. During internal validation, the ontology tagging pipeline achieved a precision of 0.92 and recall of 0.88 across a set of 500 manually annotated queries. Errors were most often associated with compound terms or overlapping entities (e.g., “MSI-H colon tumors with PTEN loss”), which were addressed by refining entity parsing rules and retraining the model with extended examples. This dual strategy balances interpretability with flexibility, enabling accurate and efficient tagging of relevant clinical-genomic concepts within natural language queries.”
Reviewer 2 writes:
- Although the statistical tests used are listed, there is no explanation of how the issue of multiple testing was addressed or whether any correction methods (e.g., FDR, Bonferroni) were applied.
Response: We appreciate the reviewer’s comment highlighting the importance of addressing multiple hypothesis testing. In response, we have added a paragraph to the manuscript clarifying how multiple comparisons were handled in this study. As the primary objective was to validate the AI-HOPE-PI3K system using previously reported associations, we did not apply correction methods in this initial evaluation phase. However, we acknowledge the importance of these techniques in discovery-focused analyses and have outlined future plans to incorporate multiple testing correction modules within the platform.
The Methods text on page 13, line 498, now reads “In this validation-focused study, statistical analyses—such as Kaplan–Meier survival modeling and odds ratio testing—were performed to replicate previously established associations in CRC, following methodologies described in our prior publication [1]. As the primary goal was to evaluate the AI-HOPE-PI3K platform’s ability to reproduce known findings using natural language–driven queries, formal correction for multiple hypothesis testing (e.g., Bonferroni or false discovery rate [FDR] adjustments) was not applied. We recognize, however, that in exploratory or high-throughput analyses involving multiple comparisons, such corrections are essential to control for type I error. Accordingly, future versions of AI-HOPE-PI3K will incorporate options for multiple testing correction, enabling users to apply standard procedures for significance adjustment during large-scale subgroup or biomarker analyses.”
Reviewer 2 writes:
- Even though analyses are described as “reproducible,” it is unclear whether the platform generates reproducible code or whether users can export full analysis protocols. It should be clarified how reproducibility is ensured—whether results are version-controlled with metadata or whether automated report files are generated.
Response: We thank the reviewer for raising this important point regarding reproducibility. To ensure transparency and reproducibility, we have released the full AI-HOPE-PI3K codebase, including the version of the platform used for the analyses described in this manuscript, on our laboratory’s GitHub repository (https://github.com/Velazquez-Villarreal-Lab/AI-PI3K) This link has been included in the Data Availability Statement for accessibility. The platform supports reproducible research practices by generating traceable query logs, annotated metadata, and structured output files that include both visualizations (e.g., survival curves, bar plots) and tabular statistical summaries. Each query submitted through the platform is associated with a time-stamped session, allowing results to be version-tracked and fully reproducible. Additionally, the system supports the export of natural language input, parsed variables, and generated code, providing users with end-to-end documentation of the analysis pipeline. We have added a paragraph to the manuscript clarifying these features and outlining our ongoing efforts to enhance reproducibility through automated report generation and integration of version control frameworks.
The abstract text on page 13, line 509, now reads “To ensure reproducibility, the AI-HOPE-PI3K platform is designed to log natural language queries, parsed parameters, and the corresponding auto-generated statistical code for each session. These elements are stored with time-stamped metadata, enabling reproducible reruns of all analyses. Users can export their query histories, cohort definitions, statistical results, and visual outputs as part of a session report. The full version of the platform, including the source code, sample queries, and validation datasets, has been released (see Data Availability Statement) to promote open science. Future updates will include automated generation of downloadable analysis reports and expanded integration with version control tools to further strengthen transparency and reproducibility.”
Reviewer 2 writes:
- The conclusion section summarizes the overall concept but is only weakly supported by the results presented in the article. As a result, the conclusions appear more promotional than analytical.
Response: We appreciate the reviewer’s thoughtful feedback and agree that the conclusion should more closely reflect the specific results presented in the manuscript. In response, we have revised the conclusion to focus on the validated performance of AI-HOPE-PI3K, its demonstrated capabilities based on the study findings, and its practical contributions to precision oncology. We have removed language that may appear overly promotional and instead emphasize the system’s current functionality, validation scope, and areas for future development in a balanced, evidence-supported manner.
The conclusions text on page 14, line 534, now reads “In conclusion, AI-HOPE-PI3K demonstrates a functional and reproducible approach to natural language–driven clinical-genomic analysis of PI3K pathway alterations in CRC. The platform successfully replicated known associations—such as site-specific survival differences and TMB-linked outcomes—and enabled exploratory analyses across ancestry, stage, and treatment-based subgroups. While several findings were hypothesis-generating rather than statistically conclusive, they highlight the platform’s utility in stratified cohort analysis and biomarker evaluation. By integrating harmonized clinical and genomic datasets with a fine-tuned biomedical LLM, AI-HOPE-PI3K lowers technical barriers to performing real-time, population-aware analyses without requiring coding expertise. As the platform continues to develop, future enhancements will focus on incorporating multivariate models, multiple testing corrections, and expanded data sources to support more comprehensive precision oncology research. Ultimately, this platform lays the foundation for an interoperable AI-agent infrastructure that lowers technical barriers to conducting pathway-specific analyses by enabling users without programming expertise to interact with complex biomedical data through natural language queries, thereby fostering inclusive, data-informed precision medicine.”
We thank the reviewer for their time, effort, and thoughtful evaluation of our manuscript. We greatly appreciate the constructive feedback and careful consideration provided, which have helped us improve the scientific clarity, rigor, and overall quality of the work. Your insights have been instrumental in guiding meaningful revisions, and we are grateful for your contribution to enhancing the value and impact of this research.

Round 2
Reviewer 2 Report
Comments and Suggestions for Authors
The authors' response demonstrates attention to the received comments and efforts to improve the manuscript.
Overall, the authors have shown an understanding of the feedback and have successfully integrated it into the revised version.
In my opinion, the manuscript has been significantly improved compared to the original version and is now close to meeting the publication requirements.